 eLIFE

# Adult-born granule cells mature through two functionally distinct states

**János Brunner[1,2†], Máté Neubrandt[1,2†], Susan Van-Weert[1], Tibor Andrási[1,2], Felix B Kleine Borgmann[3], Sebastian Jessberger[3*], János Szabadics[1*]**

[1]Lendület Laboratory of Cellular Neuropharmacology, Institute of Experimental Medicine, Hungarian Academy of Sciences, Budapest, Hungary; [2]János Szentágothai School of Neurosciences, Semmelweis University School of PhD Studies, Budapest, Hungary; [3]Brain Research Institute, Faculty of Medicine and Science, University of Zurich, Zurich, Switzerland

**Abstract** Adult-born granule cells (ABGCs) are involved in certain forms of hippocampus-dependent learning and memory. It has been proposed that young but functionally integrated ABGCs (4-weeks-old) specifically contribute to pattern separation functions of the dentate gyrus due to their heightened excitability, whereas old ABGCs (>8 weeks old) lose these capabilities. Measuring multiple cellular and integrative characteristics of 3- 10-week-old individual ABGCs, we show that ABGCs consist of two functionally distinguishable populations showing highly distinct input integration properties (one group being highly sensitive to narrow input intensity ranges while the other group linearly reports input strength) that are largely independent of the cellular age and maturation stage, suggesting that 'classmate' cells (born during the same period) can contribute to the network with fundamentally different functions. Thus, ABGCs provide two temporally overlapping but functionally distinct neuronal cell populations, adding a novel level of complexity to our understanding of how life-long neurogenesis contributes to adult brain function.

**\*For correspondence:**
jessberger@hifo.uzh.ch (SJ);
szabadics.janos@koki.mta.hu (JS)

†These authors contributed
equally to this work

**Competing interests:** The
authors declare that no
competing interests exist.

**Reviewing editor:** Gary L
Westbrook, Vollum Institute,
United States

## Introduction

Adult neurogenesis contributes to certain forms of hippocampus-dependent behavior and is associated with a number of neuro-psychiatric diseases (*Parent and Murphy, 2008*; *Deng et al., 2010*; *Kheirbek et al., 2012*; *Spalding et al., 2013*). Recent data suggested that young ABGCs (around 4 weeks old) contribute to a discrete pattern separation function, whereas older cells (8 weeks or older) are not necessary for this dentate gyrus-dependent function, therefore functionally different pools of granule cells provide unique plasticity to the hippocampal circuits (*Clelland et al., 2009*; *Aimone et al., 2010*; *Alme et al., 2010*; *Sahay et al., 2011a*; *Nakashiba et al., 2012*; *Neunuebel and Knierim, 2012*). Current theories on adult neurogenesis are based on the provisional correlations between the two distinct physiological functions and age-dependent maturation of cellular (including synaptic, biophysical and molecular) properties (*Aimone et al., 2006*, *2010*; *Sahay et al., 2011b*). This is supported by numerous observations showing that after their birth, ABGCs undergo a continuous maturation process, lasting for 8–10 weeks. ABGCs acquire neuronal properties including synaptic inputs and outputs, and capability of firing action potentials 3 to 4 weeks after their birth. Notably, ABGCs at this cellular age are highly excitable, show enhanced synaptic plasticity and are differently modulated by inhibition compared to ABGCs at the end of the maturation period (*Wang et al., 2000*; *Schmidt-Hieber et al., 2004*; *Laplagne et al., 2006*; *Toni et al., 2008*; *Mongiat et al., 2009*; *Gu et al., 2012*; *Marín-Burgin et al., 2012*; *Vivar et al., 2012*; *Dieni et al., 2013*). However, it remains unknown how two populations emerge by a continuous maturation of the underlying cellular properties of ABGCs. How do individual ABGCs transform from 'young' to 'old' properties? There are three testable

**eLife digest** Remembering what happened on different occasions involves a process in the brain called pattern separation, which allows us to separate and distinguish our memories. One part of the brain where pattern separation occurs is called the dentate gyrus, which sits in the hippocampus—the brain region that is in charge of certain forms of learning and memory.

Neurons called granule cells are thought to play a central role in hippocampal pattern separation. These cells, unlike the majority of nerve cells, can form at any time, and those that form in the mature brain are called adult born granule cells (ABGCs). Although it usually takes 10 weeks for these cells to fully mature, they are capable of communicating with each other about 3–4 weeks after being generated. Previously, it had been reported that while young, 4-week-old ABGCs are required for pattern separation, slightly older (8 week old) ABGCs are not.

What intrinsic properties make ABGCs capable of contributing to pattern separation? Is this property defined by the fate (i.e. a predetermined program) of the cell, or by the cell's experiences and activities?

To investigate these questions, Brunner et al. labeled ABGCs with a fluorescent tag when these neurons were born in adult male rats. Then, when the tagged cells were aged between 3 and 10 weeks old, the electrical properties of the labeled cells were measured from thin brain slices.

Brunner et al. found that ABGCs respond to input signals with two different levels of sensitivity. The youngest cells (3–5 weeks old) are exceptionally sensitive to a narrow range of input signal strengths, which is useful for pattern separation. The oldest investigated cells (10 weeks old), on the other hand, respond incrementally to a wide range of different input signal strengths. Under these experimental conditions, the cells changed how they respond to input signals some time between 5 and 9 weeks after being born. However, they either behaved like the youngest or like the oldest cells: no intermediate behavior was seen.

Unexpectedly, the switch is not directly related to the age of the cells: cells born at the same time don't necessarily change behavior at the same time, and cells born at different times may behave similarly. Thus, Brunner et al. suggest that it is the experience of the cells, and not their fate, that determines how they help the dentate gyrus function during the investigated period.

possibilities. First, their functionally important properties may develop continuously (*Figure 1A*). However, if this is the case, it may contradict the general notion that two distinct ABGC populations exist, and ABGCs would provide a functional continuum. The second possibility is that, as during their early maturation when becoming functionally integrated (<4 weeks), ABGCs switch function according to a predetermined program (*Figure 1B*). In this situation there are two clearly distinct populations and they would switch within a short temporal window at a predefined stage of their postmitotic life. Third, ABGCs may be susceptible to extrinsic cues allowing for a functional switch for an extended period (*Figure 1C*). We tested these hypotheses by resolving the integrative properties of individual ABGCs because data-pooling could mask the differences between above hypotheses. Thus, we here analyzed if (i) all cellular properties develop concomitantly, (ii) if there are biophysical properties that allow the emergence of only two populations between 3 and 10 weeks after their birth, and (iii) how and when ABGCs switch function. Using this approach, we show that ABGCs consist of two functionally distinct populations during an extended period, between 3 and 9 weeks of age in rats, by being sensitive to distinct aspects of their inputs.

## Results

To analyze the cellular maturation of ABGCs, we compared a variety of intrinsic biophysical and input–output transformation properties at seven different age-groups (3–10 weeks after cells are born) of individual birth-dated ABGCs in young adult rats using retroviral labeling (*Figure 2*, *Zhao et al., 2006*). The majority of the tested parameters (including input resistance, membrane time constant, whole-cell capacitance, resting membrane potential, action potential threshold, peak $dV/dt$ of the spikes, and maximal firing rate) of individual ABGCs changed continuously with age and, consequently, the distribution of the data points from individual cells was wide, without the emergence of distinguishable populations (*Figure 3A–B*, statistical values are indicated in the figures—see also *Supplementary file 1*),

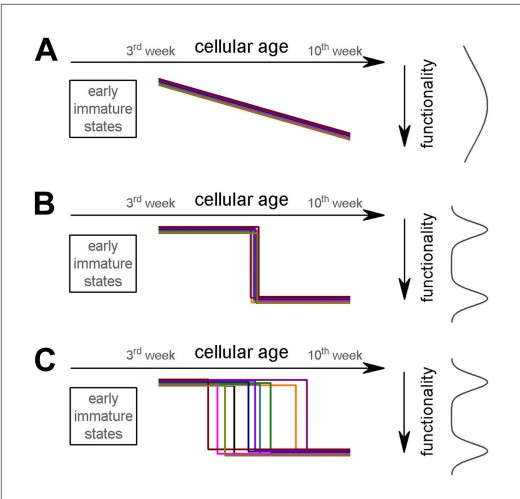

**Figure 1**. Potential theoretical modes of postmitotic maturation of functional properties. Each colored line represents the age-dependent change of a theoretical parameter from individual ABGCs. (**A**) Gradual maturation of the properties results in widely distributed functional continuum. (**B**) Temporally predefined functional switch. (**C**) The functional switch occurs in an extended temporal window.

reflecting the continuous maturation of these properties in accordance with previous observations (*Mongiat et al., 2009*; *Marín-Burgin et al., 2012*).

In addition to these basic biophysical parameters, we measured the suprathreshold input–output functions in response to sinusoidal current injections to mimic temporally organized input patterns in physiologically relevant frequency ranges (*Figure 2F–H*, *Pernia-Andrade and Jonas, 2014*). During these protocols, due to the fluctuating membrane potential, the contribution of voltage-gated ionic channels to the firing is more relevant than in the case of square pulse injection. In contrast to basic membrane properties, the analysis of the gain of the input–output functions of the same cells revealed two significantly distinct populations using Gaussian fits and K-means analysis (*Figure 3C*). The input–output function of the first group was characterized by a steep average slope (ASL) and highly variable (measured as the variance of the slope, VAR; *Figure 2F–H*) spike responses, suggesting that this cell population is exceptionally sensitive to certain narrow input strengths, and thus highly suited for disambiguating input–output functions at single cell level (notice the out-of-average values on *Figure 2H* for the first and third cells, hereafter

referred to as S-group, standing for *Sensitive*). Within the S-group the integrative parameters were independent of the actual age of the individual cells (linear fits on *Figure 3C*) demonstrating that similar cellular functionality is maintained throughout an extended period (between 3–9 weeks) unless the individual cell switched to the second integration mode. This second group of ABGCs responded with constantly and incrementally increasing, less sensitive spike output (L-group; referring to *Linear*) enabling them to linearly report various input strengths. A different parameter of the input–output functions, the offset did not divide the same ABGC samples into two populations, indicating that the functional separation of ABGCs is restricted to specific cellular properties (*Figure 3B*). Altogether, the above results show that ABGCs form two functionally distinct populations during a long period of their maturation based on their different sensitivity to temporally organized inputs.

The above analysis suggested that age alone does not directly determine the functionality of individual ABGCs. However, the probability of whether an individual cell behaved as a member of S- or L-groups shows age-dependence (*Figure 4A*). Within all analyzed cells, the youngest (3 weeks old) and the oldest (10 weeks old) belonged to the S- or L-group, respectively; however, both functional groups were present with changing probabilities during the intermediate ages between 5 and 9 weeks of age. This observation was further supported by the parabolic distributions of the mean-variance plots of ASL and VAR (*Figure 4B*, *Supplementary file 1*). Thus, the continuously increasing probability of L-groups could mask the two functionally distinct groups of ABGCs when global population properties were analyzed as averages (*Mongiat et al., 2009*).

Our data indicate that in 3–9 weeks old ABGCs the gain of the input–output properties is not exclusively determined by the input resistance. ABGCs within the S-group achieved similar input–output computations in spite of largely different input resistances. Strikingly, this becomes clear by the lack of correlations of the gain of individual S-group cells to their input resistance (*Figure 4C*). However, the input–output function of ABGCs from the L-group showed the expected dependence on the input resistance of individual cells. In contrast to the gain, the offset of the input–output function of individual ABGCs showed a clear dependence on the input resistance across both cell populations (*Figure 4D*). Thus, the independence of the gain of the input–output transformation from the continuously developing biophysical parameters at the level of individual cells allows for the emergence of only two temporally overlapping and functionally distinct populations within 5- to 9-weeks-old ABGCs.

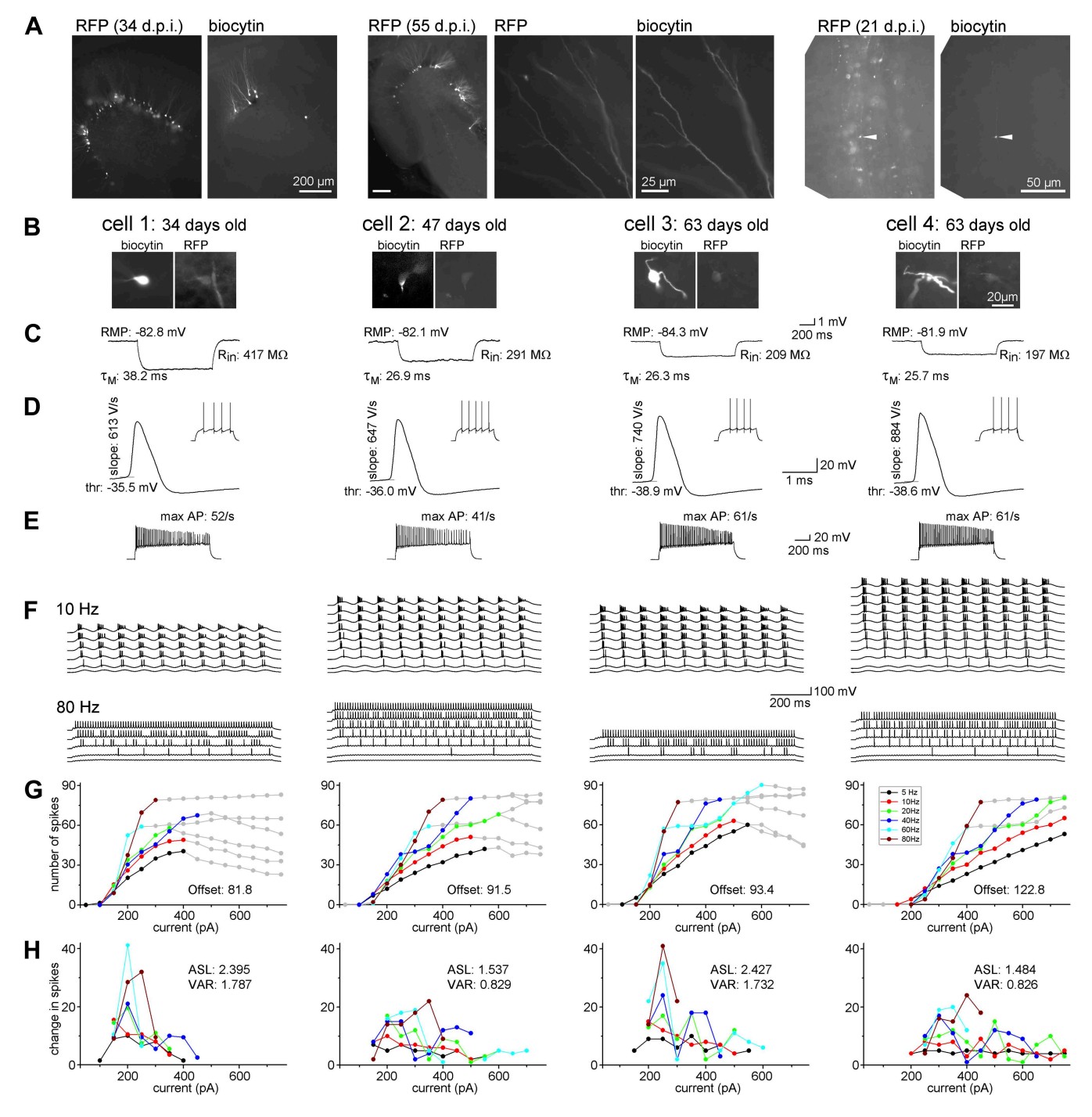

**Figure 2**. Maturation of the biophysical and integrative properties of ABGCs. (**A**) The RFP and biocytin-labeled cells in the dentate gyrus (left panels, d.p.i.: day after virus injection), spiny dendrites (middle panels), and typical mossy fiber terminals in the stratum lucidum of the CA3 region (right) confirm granule cell identity. (**B**) Four representative RFP-expressing granule cells 34, 47, and 63 days after CAG-RFP virus labeling. The 63-day-old AGBCs were recorded from the same slice. (**C**) Average subthreshold voltage responses of the example cells to small (−10 pA) current steps. Input resistance ($R_{in}$), membrane time constant ($\tau_M$), and resting membrane potential (RMP) of the cells are indicated. (**D**) Spike parameters of the example cells at lower current intensities ($dV/dt$: maximal rate of rise, thr: action potential threshold). (**E**) Maximal firing rate of the four cells in response to square pulse current injection. (**F**) Responses of the cells to sinusoidal current injections with increasing amplitude (Δ50 pA) at 10 and 80 Hz. The traces are shown until the firing reached saturation. (**G**) Number of spikes generated in the example cells as a function of the peak amplitude of the injected sinusoid currents at the all tested frequencies. Gray symbols indicate values that were omitted from the analysis due to lack or saturation of spiking. Offset values describe

*Figure 2. Continued on next page*

*Figure 2. Continued*

the minimum input intensities to reach 50% spiking output. (**H**) Increments of the firing (i.e., the first derivative of the curves in panel **F**) of the cells. These values were used for the calculation of the average slope (as mean, ASL) and the variance of firing (as variance, VAR). Note that cells 1 and 3 have exceptionally large values at certain input intensity ranges indicating that these cells were more sensitive to certain input intensities. This characteristic is quantified by the large VAR value.

Importantly, independent experiments in which granule cells were recorded in non-labeled adult animals ('Materials and methods'), confirmed the coexistence of S- and L-functionalities among granule cells based on their ASL and VAR, and these individual cells showed similar correlations (or lack thereof) between their integrative functions and input resistance (*Figure 4—figure supplement 1*) as in the case of the above birth-dated data set (*Figure 4C*).

Next, we tested whether all measured biophysical parameters of individual birth-dated ABGCs can cooperatively predict the functional separation at the level of the gain of their input–output functions. We performed cluster analysis of the recorded cells based on seven intrinsic parameters to define two groups (*Figure 4E*). The two intrinsic parameter groups determined by cluster analysis did not match with the S- and L-group identity of the same individual cells. This latter result shows that consideration of multiple parameters by their arithmetical values is not sufficient to predict the functional separation of S- and L-groups. Thus, a complex and balanced interaction is probably behind the formation of the two states, as also suggested by additional experiments, in which the cellular excitability was altered by decreasing the temperature, adding extracellular calcium or activating background conductances (*Figure 4—figure supplement 2A–C*). We also tested whether the two distinct input–output functions are due to their distinct synaptic drives (*Dieni et al., 2013*) by blocking GABA$_A$ and AMPA/kainate receptors. This intervention slightly increased the ASL value (*Figure 4—figure supplement 2D*); however, the effect was not dependent on the initial state of the tested cells indicating that S- and L-group properties are established largely independent of spontaneous synaptic activity.

## Discussion

Here, we show that the gain of the input–output transformation of 3- to 10-weeks-old ABGCs exists at two functionally distinct states, allowing for the translation of similar excitatory drives into highly distinct action potential outputs in a manner that is not directly predicted by the cellular age alone (refer to the alternative maturation hypotheses depicted in *Figure 1A–C*). This finding is in contrast to the continuous maturation hypothesis (*Figure 1A*). Around the third postmitotic week, ABGCs represent a functionally homogeneous population (S-group) characterized by highly variable and sensitive output, which potentially underlies the effective disambiguation of input patterns because the output of S-group cells represent certain input ranges with exceptional efficacy. Importantly, this functional parameter is similar for an extended time period at the level of individual ABGCs, despite the continuous maturation of other biophysical parameters suggesting precise homeostatic tuning and complex interactions of the biophysical properties (*Marder and Goaillard, 2006*). However, between the fifth and ninth week (under our experimental conditions), ABGCs switch function by losing their sensitivity to a particular input strength as their output incrementally reports a wide input range (L-group). Thus, in the theoretical case of identical input strengths and patterns to S- and L-cells (which allowed us to investigate the integrative properties of individual ABGCs in isolation), sensitivity of ABGCs in the S-group to certain input ranges, in opposed to linear reporting in cells of the L-group, allows a certain level of input disambiguation at the level of single granule cells. Furthermore, previously described distinct input rules (*Dieni et al., 2013*) may synergistically promote two distinct functions within the ABGC populations, if these rules are specifically associated with S- or L-group properties.

Strikingly, our data indicate that 'classmate' cells (born during the same period) can contribute to the network with fundamentally different functions during an extended period after cells are born (5–9 weeks). Conversely, these data suggest that similar cellular functions can be served by ABGCs that were born at different periods during the animal's life. Therefore, the properties of individual cell rather than the cells' age determine how certain input strengths are computed; either by, disambiguating certain input combinations (S-functionality) or by representing all of its inputs by similar rate changes toward the downstream networks (L-functionality). This observation challenges previous hypotheses on the plasticity provided by adult neurogenesis from being predominantly determined by

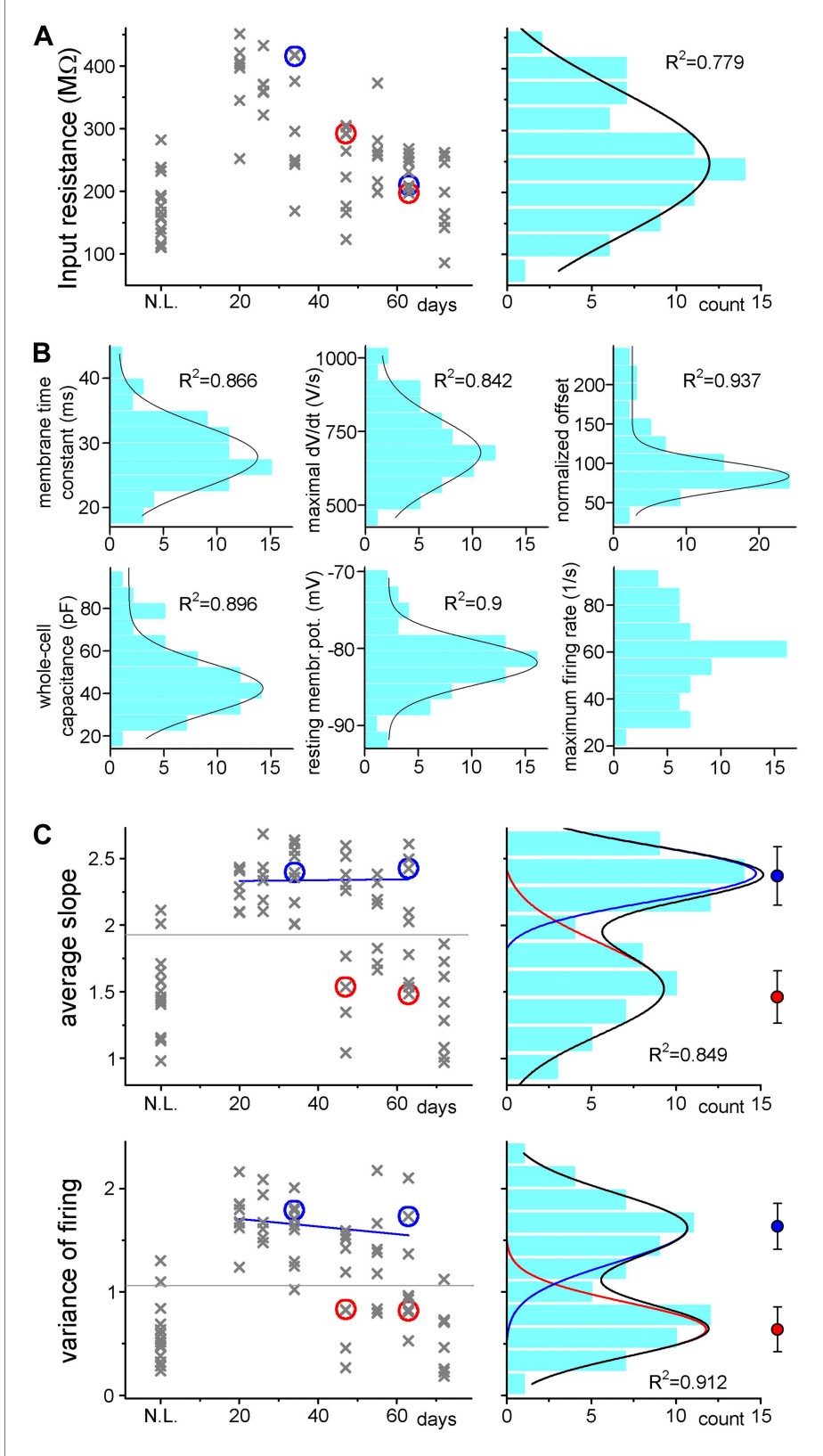

**Figure 3**. Adult born granule cells (3–10 week old) can be divided into two distinct populations based on cell-to-cell differences in input–output transformation. (**A**) Left, input resistance of individual ABGCs with various ages (gray crosses). Red and blue circles (S- and L-group members, respectively) highlight the values for the example

*Figure 3. Continued on next page*

*Figure 3. Continued*

cells shown in *Figure 1*. N.L.: not labeled control cells. Right, probability distribution of the data set shows single peak (single Gaussian fit: F = 0.0001). (**B**) Monotonous probability distribution of membrane time constant, whole cell capacitance, resting membrane potential, maximal rate of rise of spikes, relative offset of the input–output curves, and action potential threshold data from the same set of cells as above. (**C**) Left, average slope (top) and variance of the slope (bottom) of the same individual ABGCs as above with various ages (gray crosses). Blue lines indicate the lack of correlation between the gain of the input–output functions and the age of individual cells within the S-group (linear fit, ASL: $R^2 = -0.029$, p = 0.89, VAR: $R^2 = 0.01$, p = 0.257). Right, two population emerges from the distribution of the average slope values of individual cells (two peaks Gaussian, ASL: F = 0.0014, VAR: F = 0.0001). The centers of the two clusters and average distance values from the centers (error bars) are shown on the right (K-means analysis, $F < 10^{-9}$, horizontal gray lines on the left panels indicate the separation by the K-means analysis).

the postmitotic cellular age and predicting similar functions of ABGCs born at the same time (*Aimone et al., 2006*, *2010*; *Sahay et al., 2011b*). Moreover, environmental conditions that increase or reduce hippocampal neurogenesis may affect the relative contribution of newly generated granule cells to the S- or L-groups that may explain distinct behavioral consequences of altered neurogenesis (*Kempermann et al., 1997*; *Gould and Tanapat, 1999*; *van Praag et al., 1999*).

## Materials and methods

All experimental procedures were made in accordance with the ethical guidelines of the Institute of Experimental Medicine Protection of Research Subjects Committee (permission: 22.1/1760/003/2009) and were approved by the local virus safety committee.

### Virus mediated birth-dating of granule cells

31- to 33-day-old male Wistar rats (95–135 g body weight) were injected with a CAG-GFP or CAG-RFP Moloney murine leukemia virus vector (*Zhao et al., 2006*; *Jessberger et al., 2007*) (0.8–1 µl) using stereotaxically targeted (5.7–5.8 mm posterior, ± 4.4–4.5 mm lateral, and 5.6–6 mm ventral from bregma), conventional Hamilton syringe under ketamine/xylazine/pipolphen anesthesia (83/17/7 mg/ body kg). Adult born granule cells were labeled along a broad longitudinal range (2–3 mm) of the hippocampi. Note that we did not find cells in animals 3–10 weeks after virus injection, which had <3-week-old properties indicating the reliability and precision of the birth-dating method. Animals of this age were used because relatively large number of labeled ABGCs can be analyzed at a time of recording when the network properties of the dentate gyrus circuitry can be considered adult (*Laplagne et al., 2006*). After the surgical procedure two or three siblings were housed together in large cages (75 cm × 35 cm) equipped with a running wheel, toys and shelters until the electrophysiological experiments because running is known to increase the number of surviving adult-born neurons (*Kempermann et al., 1997*; *Tashiro et al., 2007*; *Dranovsky et al., 2011*). For acute slice preparations, rats were deeply anaesthetized with isoflurane and slices (300–350 µm) were cut in ice-cold artificial cerebrospinal fluid (consisting of 85 mM NaCl, 75 mM sucrose, 2.5 mM KCl, 25 mM glucose, 1.25 mM $NaH_2PO_4$, 4 mM $MgCl_2$, 0.5 mM $CaCl_2$, and 24 mM $NaHCO_3$). The orientation of cutting was perpendicular to the axis of the hippocampus at the level of virus injection. After the cutting, slices were kept at 32°C for 30 min and then at room temperature.

### Electrophysiological recordings

During the recordings, slices were perfused with a solution containing 126 mM NaCl, 2.5 mM KCl, 26 mM $NaHCO_3$, 2 mM $CaCl_2$, 2 mM $MgCl_2$, 1.25 mM $NaH_2PO_4$, and 10 mM glucose, 35–36°C. To reduce the instrumental capacitance (including pipette capacitance), recording pipettes were pulled from thick glass (i.d. 0.87; o.d. 1.5 mm). Pipette resistance was in the range of 5.5–9.5 MΩ, and the usual total instrumental capacitance was 9.5–11 pF, which was neutralized to the maximum obtainable level (<2 pF remaining capacitance) under current clamp conditions (digitized at 50 kHz and low-pass filtered at 20 kHz). The intracellular solution contained 90 mM K-gluconate, 43.5 mM KCl, 1.8 mM NaCl, 1.7 mM $MgCl_2$, 50 µM EGTA, 10 mM HEPES, 2 mM Mg-ATP, 0.4 mM $Na_2$-GTP, 10 mM phosphocreatine-disodium, and 8 mM biocytin (pH 7.25). Note that because of the relatively high chloride concentration in the intracellular solutions, differences in the cellular properties are unlikely due to

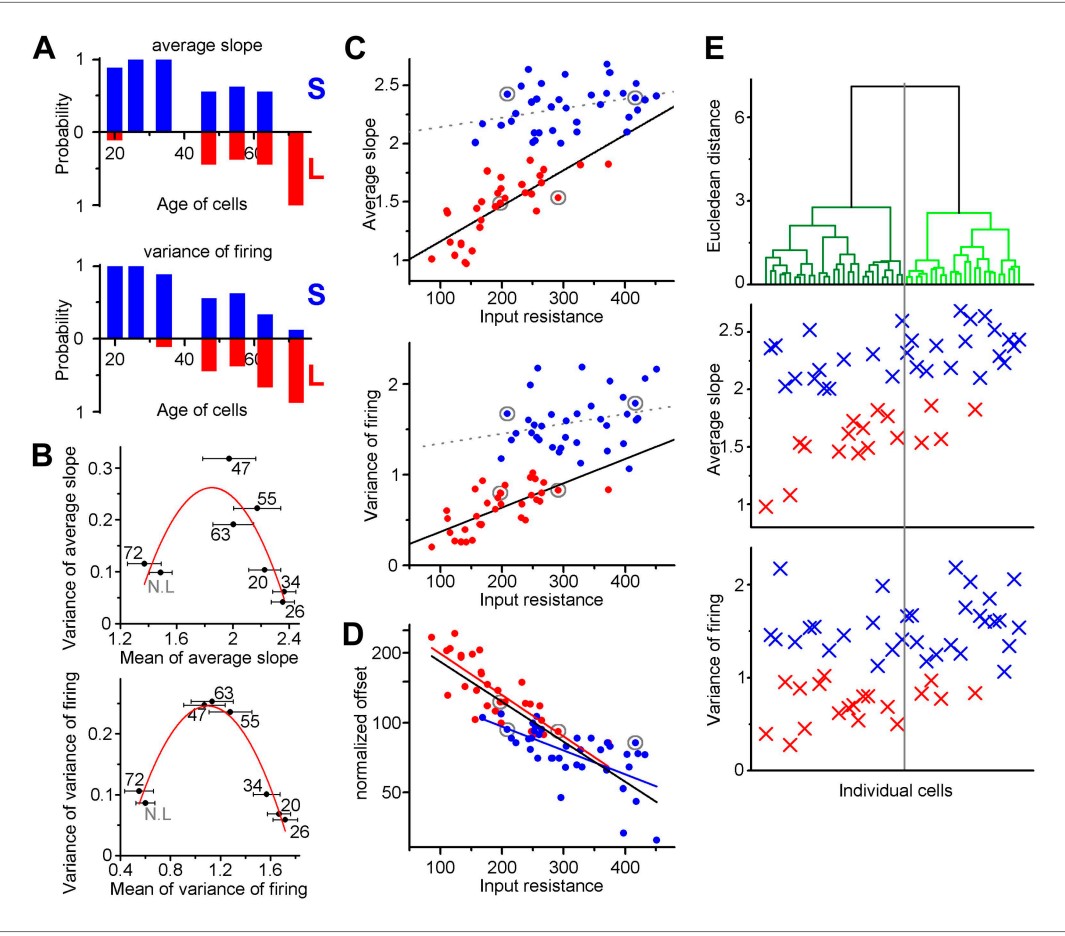

**Figure 4**. Independence of the output properties of individual ABGCs from age and input resistance. (**A**) Probability of the members of the clusters defined by the K-means analysis continuously shifts from S-group (blue) toward L-group (red) during maturation indicating the higher prevalence of ABGCs with shallow and invariable input–output function. (**B**) The population level functional switch is also suggested by the higher variance of the input–output parameters during the transition age period and low variance in the youngest and most matured populations (parabolic fit, ASL: $R^2 = 0.611$, F(ANOVA) = 0.0035; VAR: $R^2 = 0.853$, F = 0.00017). Numbers indicate the age of the data sets. (**C**) Correlation of the integrative parameters to the input resistance within the two functionally different groups (red and blue symbols) defined by K-means cluster analysis (see ***Figure 3***). Gray circles indicate the four example cells from ***Figure 2*** (linear fits, ASL: $R^2 = 0.087$, $p = 0.045$ for S-group, $R^2 = 0.576$, $p = 2.9 \times 10^{-7}$ for L-group; VAR: $R^2 = 0.02$, $p = 0.19$ for S-group $R^2 = 0.467$, $p = 2.6 \times 10^{-6}$ for L-group). (**D**) Correlation between the normalized current needed to reach output that is half of the input frequency and input resistance of individual ABGCs belonging to the two functionally different groups ($R^2 = 0.39$, $p = 0.0001$ for S-group; $R^2 = 0.607$, $p = 2 \times 10^{-8}$ for L-group; $R^2 = 0.682$, $p < 10^{-8}$ for both groups). (**E**) Considering multiple membrane parameters of the individual cells (resting membrane potential, membrane time constant, whole cell capacitance, input resistance, threshold, peak $dV/dt$, maximal firing rate) for hierarchical cluster analysis (Ward method with normalized values) were not sufficient to predict the functional identity of the cells. Data from individual cells are aligned vertically (lines in the upper panel and symbols in the middle and bottom panels). Thus, the order of the points along the X-axes is determined by the results of the cluster analysis.

The following figure supplements are available for figure 4:

**Figure supplement 1**. Coexistence of S- and L-functionalities among granule cells from non-labeled animals.

**Figure supplement 2**. The input–output transformation of individual cells is maintained in two stable states by complex mechanisms.

age-, stage-, maturation-level-type specific chloride homeostasis (*Overstreet-Wadiche and Westbrook, 2006*; *Markwardt et al., 2009, 2011*). Note that the properties were tested while the spontaneous synaptic activity was left intact (i.e., no blockers were included during control conditions).

The expression of GFP or RFP was verified usually by multiple criteria: match between the epifluorescence (excitation at 490–510/540–580 nm, detection at 520LP/593–667 nm for GFP/RFP) and Nomarski (900 nm) differential interference contrast images (Eclipse FN-1; Nikon, Japan), appearance of fluorescent signal in the recording pipette and washout of the intracellular labeling during the recording, and post hoc colocalization of fluorescent intracellular biocytin–and RFP/GFP signals. The epifluorescent illumination of slices was reduced as much as possible before and during the recordings in order to avoid any photo-damage of the labeled cells and usually did not last longer than few seconds above the cells. In majority of slices, in addition to birth-dated ABGCs, we also recorded non-labeled control cells, which located on the border of strata granulosum and moleculare, in order to provide controls for similar recording conditions across animals. Note that because virus injection took place always in P31–33 animals, their age at the time of recordings varied between P51 and P105 (corresponding to recording of 20–72 days old ABGCs). Notably, no correlations were found between the age of the animals and cellular properties of non-labeled control cells. Semilunar granule cells, characterized by extremely low (<90 MΩ) input resistance and broad dendritic arbor, were excluded from the analysis.

For post hoc anatomical processing, slices were fixed for a day in 0.1 M phosphate buffer containing 2% paraformaldehyde and 0.1% picric acid at 4°C. For visualizing the biocytin signal, sections were incubated overnight with Alexa Fluor 350-conjugated streptavidin (1:500; Invitrogen, Carlsbad, CA) in 0.5% Triton X-100 and 2% NGS containing TBS buffer at 4°C. After washing and mounting in Vectashield (Vector Laboratories, Burlingame, CA), the endogenous signal of the fluorescent protein was compared with the biocytin staining by using epifluorescent illumination (DM2500; Leica, Germany).

## Characterization of the integrative and biophysical properties of individual ABGCs

To reliably determine the potential correlations between the different intrinsic parameters, we collected data for each tested parameters from each analyzed cells (3–10 weeks old). ABGCs in the early phase of their maturation (younger than 3 weeks) were not analyzed because they are not yet fully integrated into the hippocampal network due to the lack of reliable high frequency spiking, which is the consequence of lower sodium channel densities. In order to measure the input–output characteristics of ABGCs during different stages of their maturation, we used sinusoidal current injection from theta to high gamma frequency bands (5, 10, 20, 40, 60, 80 Hz, 50 pA increment from the holding level of 0 pA, tested in random order) for 1 s and analyzed the number of the elicited action potentials. Application of sinusoidal current injections from resting membrane potential mimicked temporally correlated excitatory drives in functionally relevant frequency ranges, in opposed to square pulses, which would strongly recruit non-physiological mechanisms such as inactivation of voltage activated conductances. This type of mimicking of the excitatory drive to GCs is justified by the single supralinear integration zone of GCs (i.e., spike initiation in the axon initial segment [*Krueppel et al., 2011*]). Furthermore, it has been reported that the amplitude of miniature excitatory events recorded at the somata does not increase further after the 3–4 weeks of ABGCs (*Mongiat et al., 2009*). Using somatic current injection, thus, avoids the potential confounds introduced by short-term plasticity upon repetitive stimulation of the same fibers.

We characterized the integrative properties of individual ABGCs by two reliable parameters, which measure the gain of the input–output functions: the average slope (ASL) and the variance (VAR) of the slope of input–output curves. The calculated parameters (average slope, variance, and offset) were weighted by the different frequencies using empirically determined correlations to obtain a pooled, frequency-independent data point from each recorded cells. Average slope (ASL) was calculated as the arithmetic mean of the first derivative of the input–output function and weighted by the square root of the frequency. Frequency-weighted variance of the gain of the firing (VAR) was calculated as the variance of the first derivative of the input–output function divided by the frequency. Thus, these two measures are sensitive to different aspects of the input–output function of a given cell and characterize individual cells with a single and reliable value. High ASL value suggests that the given cell is capable of large output changes in response to unitary input changes, whereas the large VAR highlights that the cell is more sensitive to a particular input intensity range. Importantly, the ASL and VAR values remained stable for individual cells provided stable membrane potential, input resistance, and

capacitances values and well-compensated bridge balance. These parameters were strictly monitored in every recorded trace using a 50 ms long −50 pA step and manually corrected if necessary and recordings were excluded if the resting membrane potential changed more than 4 mV compared to the initially measured values.

Input resistance was measured as the average steady-state voltage response to −10 pA current steps (30–100 traces excluding traces with large spontaneous events). Membrane time constant was fitted with single exponential on these traces between 2–100 ms both at the onset and the end of the current step. The maximum rate of rise (peak $dV/dt$) was measured on the first spike that was elicited using square pulse currents without post hoc filtering. Action potentials were defined as larger deflection in the first derivative of the recorded voltage trace than 20 mV/ms following post hoc low-pass filtering at 4 kHz. The maximum firing capability of the cells was challenged by 1 s long square current injections with increasing amplitude (Δ20 pA) until depolarization block was reached. Action potential threshold was measured as the voltage at 20 mV/ms of the $dV/dt$. The whole cell capacitance was measured in voltage clamp recordings using a −5 mV voltage step at −70 mV holding by measuring the integral area of the current response (measured from the steady-state current level) and divided by the voltage step amplitude. The offset of the input–output function was defined as the peak amplitude of the current waveform necessary to reach larger firing frequency than the half of the input frequency. For normalization, we weighted the values with the fourth root of the input frequencies. Correlations are characterized with adjusted R-square ($R^2$).

In an independent subset of experiments to provide evidence for the existence of S- and L-functionalities to test the potential underlying mechanisms, the properties of granule cells were tested in animals, which were not subject to virus injection. These were adult rats (P68–101) kept with running wheels (*Figure 4— figure supplements 1–2*), and the granule cells were recorded mostly in the lower half of the cell layer. The same criteria were applied for these cells as in the case of the birth-dated ABGCs.

## Acknowledgements

This work was funded by the Wellcome Trust (International Senior Research Fellowship #087497 to JS), the Hungarian Academy of Sciences (Lendület Initiative #LP-2009–009 to JS), Gedeon Richter (to JS), the Swiss National Science Foundation (to SJ), and the EMBO Young Investigator program (to SJ). We thank Alejandro Schinder, László Acsády, and Simon MG Braun for comments on the manuscript. We thank the Nikon Microscopy Center of Excellence at IEM for providing microscopy support, László Barna for his help with the statistical analysis and imaging, Dóra Kókay, Endre Marosi, Andrea Juszel, and Dóra Hegedűs for technical assistance. FBKM current address is Luxembourg Centre for Systems Biomedicine, Université du Luxembourg.

## Additional information

### Funding

| Funder | Grant reference number | Author |
| --- | --- | --- |
| Wellcome Trust | 087497 | János Szabadics |
| Magyar Tudományos Akadémia | Lendulet LP-2009-009 | János Szabadics |
| Swiss National Science Foundation | | Sebastian Jessberger |
| European Molecular Biology Organization | | Sebastian Jessberger |
| Gedeon Richter | | János Szabadics |

The funders had no role in study design, data collection and interpretation, or the decision to submit the work for publication.

### Author contributions

JB, MN, Acquisition of data, Analysis and interpretation of data, Drafting or revising the article; SV-W, TA, FBKB, Drafting or revising the article, Contributed unpublished essential data or reagents; SJ, Conception and design, Drafting or revising the article; JS, Conception and design, Acquisition of data, Analysis and interpretation of data, Drafting or revising the article

## Ethics

Animal experimentation: All experimental procedures were performed in accordance with the ethical guidelines of the Institute of Experimental Medicine Protection of Research Subjects Committee (permission: 22.1/1760/003/2009) and were approved by the local virus safety committee.

## Additional files

### Supplementary file

• Supplementary file 1. Measured parameters from individual ABGCs—*Brunner Neubrandt data ABGC.xlsx*.

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
