## [Decision Letter]

Thank you for sending your work entitled “Adult-born granule cells mature through two functionally distinct and stable stages” for consideration at *eLife*. Your article has been favorably evaluated by Eve Marder (Senior editor) and 2 reviewers as well as a member of our Board of Reviewing Editors. The Reviewing editor and the other reviewers discussed their comments before we reached this decision, and the Reviewing editor has assembled the following comments to help you prepare a revised submission.

The authors of this manuscript present an interesting analysis of the intrinsic input-output properties of adult born neurons across at various developmental stages. They make two conclusions. First, they show that ABGCs display heterogeneous input-output characteristics that suggest the existence of two functional populations with distinct intrinsic integrative capabilities. Second, they show that these populations are not strictly dictated by the post mitotic age of the cell, although the probability shifts with age of the cell. The authors demonstrate the emergence of two distinct groups (“R” or “K”) independent of cellular age and intrinsic membrane properties. Whereas most adult born dentate granule cells that are 3-5 weeks of age belong to the R group (highly sensitive and variable input-output transformation), adult born dentate granule cells 9 weeks and older belong to the K group (incremental, non variable response to differing input strengths). Interestingly, 5-9 weeks old adult born dg neurons may be either R or K group members. The conclusion that adult-born neurons can display heterogeneous intrinsic properties not determined by the age of the cell is a potentially important observation that challenges the widespread assumption that the function of ABGCs depends on post mitotic cell age. However, the essence of the conclusions are based on a difference in only 1 (input-output transformation) of the 7 (mostly passive properties) measured. Without further demonstration or explanation of the underlying mechanism, it is difficult to be certain of the strong conclusions put forward by the authors without further revision.

Substantive comments:

1) The authors make the point that input-output properties of ABGCs are not determined exclusively by input resistance. A prior study also reported heterogeneous intrinsic excitability of ∼4 week old cells (Dieni et al., J. Neurosci. 2014). The authors should discuss this prior work. In particular, the authors should explain why their analysis with sine wave stimuli reveals different features than are seen with conventional input resistance measurements. For example, does the sine wave activate specific voltage-gated channels? See also point 3.

2) The Methods do not mention inclusion of receptor antagonists during recordings, so it is not clear whether differences in spontaneous synaptic activity between cells could contribute to the reported differences (assumed to be intrinsic). The authors should state whether antagonists were used in the Methods, and if not, confirm that each population persists when receptor antagonists are present.

3) It is surprising that the authors did not address the mechanism of differences in intrinsic excitability. Although the identification and time-dependence of the two populations of cells is interesting, the experimental results seem thin without experimentally addressing the source of the difference. The manuscript consists of a single data set (Figure 2) that has been extensively analyzed (Figures 3 and 4). Addressing the source of the presumably intrinsic difference would increase the significance of the findings, especially when there are obvious candidates (e.g. Ca^2+^ channels, Schmidt-Hieber et al., Nature 2004).

4) The authors assert that the two types of input-output transformations differentially support pattern separation at the cellular level. The idea that pattern separation (a network function) can occur on a single cell level requires further explanation or justification.

5) The authors used running wheels/exercise that has been shown to accelerate rate of maturation of adult-born dg neurons. As such, this experimental design confounds the extrapolation of “stage specific” properties at baseline. The authors must explicitly acknowledge this potential confound and the possibility that the window of the R and K stages may actually be shifted farther to the right in the lineage at baseline (without exercise induced acceleration of maturation). How do the authors rule out the possibility that the distribution of groups (R, K) may be due to the differential effects of modulatory afferents onto maturing ABGCs by running?

6) It is not clear why such young rats were used for this study (besides the fact that this age may be more amenable for retroviral injections), if the authors were interested in studying adult hippocampal neurogenesis? Do we know that the same distributions of ABGCs will be found in adult brains?

7) Because 5-9 weeks old neurons are made up of R and K like cells, how might this mixed population contribute to encoding functions (pattern separation and completion) of the DG. In other words, a greater discussion on the significance of variable I-O relationships in classmate cells to encoding functions is warranted.

8) The Discussion section is confusing and needs revision to accurately reflect the actual findings of the experiments. For example, the last two paragraphs of the Discussion are almost disconnected. In paragraph 1, the authors indicate that cells in the third post–mitotic week are “functionally homogeneous”, then in the next paragraph they indicate that “classmate” cells contribute to the network with “functionally different functions”.

9) The word “stable” in the title does not seem to be supported by the data.

---

## [Author Response]

*1) The authors make the point that input-output properties of ABGCs are not determined exclusively by input resistance. A prior study also reported heterogeneous intrinsic excitability of ∼4 week old cells (Dieni et al., J. Neurosci. 2014). The authors should discuss this prior work. In particular, the authors should explain why their analysis with sine wave stimuli reveals different features than are seen with conventional input resistance measurements. For example, does the sine wave activate specific voltage-gated channels? See also point 3*.

We agree with the reviewers and have now highlighted the important work of Dieni et al. both in the Results and in Discussion. There are some crucial similarities in the findings of [6] paper and our current study, such as the unusual independence of the studied spiking properties from the input resistance. On the other hand there are substantial differences between the two studies. First, they showed distinction based on the inputs to ABGCs, whereas our study revealed a functional separation within ABGCs solely based on their intrinsic integrative properties. Furthermore, our study highlights that two functionally states temporally overlap during an extended period of the post–mitotic age of ABGCs. We are convinced that this novel finding has fundamental consequences on the potential physiological roles of adult neurogenesis. Nevertheless, in the future when more effective techniques will become available, it will be tested whether the functional separation of individual ABGCs at the levels of inputs and intrinsic properties are associated (see also the next point). Please refer to our modified Discussion section, which reads as follows: “Furthermore, previously described distinct input rules (6) may synergistically promote two distinct functions within the ABGC populations, if these rules are specifically associated with R- or K-group properties.”

*sine wave:* We have now included a more detailed justification for the *sine-wave* current stimulation of the cells both in the Results and Methods. The modified sections read as follow:

“During these protocols, due to the fluctuating membrane potential, the contribution of voltage-gated ionic channels to the firing is more relevant than in the case of square pulse injection.”

“Using somatic current injection, thus, avoids the potential confounds introduced by short-term plasticity upon repetitive stimulation of the same fibers.”

*2) The Methods do not mention inclusion of receptor antagonists during recordings, so it is not clear whether differences in spontaneous synaptic activity between cells could contribute to the reported differences (assumed to be intrinsic). The authors should state whether antagonists were used in the Methods, and if not, confirm that each population persists when receptor antagonists are present*.

The original experiments did not include any antagonists in order to assess the properties of individual ABGCs as close to their natural behavior as possible. We revised the Methods section to explicitly state this. Importantly, we have now conducted additional experiments to test the potential contribution of spontaneous synaptic activity (see Figure 4—figure supplement 2). The Results show that blockade of GABAA and AMPA/kainate receptors slightly changed the slope and the variance of the input-output function of the cells relative to control conditions in the same cells. However, this small effect was not depended on the initial values of these parameters, so R- and K-group cells were affected similarly, indicating that the spontaneous synaptic activity does not contribute to the separation between R- and K-functionalities (see also point 3 for positive controls). The new section in the Results reads as follows:

“We also tested whether the two distinct input-output functions are due to their distinct synaptic drives (6) by blocking GABAA and AMPA/kainate receptors. This intervention slightly increased the ASL value (Figure 4—figure supplement 2); however, the effect was not dependent on the initial state of the tested cells indicating that R- and K-group properties are established largely independent of spontaneous synaptic activity.”

*3) It is surprising that the authors did not address the mechanism of differences in intrinsic excitability. Although the identification and time-dependence of the two populations of cells is interesting, the experimental results seem thin without experimentally addressing the source of the difference. The manuscript consists of a single data set (*Figure 2*) that has been extensively analyzed (*Figures 3 and 4*). Addressing the source of the presumably intrinsic difference would increase the significance of the findings, especially when there are obvious candidates (e.g. Ca*^*2+*^
*channels, Schmidt-Hieber et al., Nature 2004)*.

Concerns regarding the “single data set”: We understand the reviewers’ concern and performed new experiments to provide independent data sets for the existence of R- and K-functionality among granule cells (Figure 4—figure supplement 2). Note that these were wild-type animals without virus-labeling and the properties of randomly selected granule cells in the lower half of the cell-layer were tested. We used the same criteria for this analysis as before (e.g. semilunar granule cells and cells without reliable high frequency firing were excluded). The new recordings revealed R- and K-group cells in adult rats based on the average slope and the variance of the slope. Thus, the results of this independent sample provide further support for the coexistence of R- and K-functionalities among granule cells. Importantly, the integrative functions of the cells also showed similar correlations (or lack thereof) to the input resistance as in the case of the original data set (compare Figure 4 and Figure 4—figure supplement 1). The incidences of the members of the two groups are different in the two sets of experiments; however, these approaches do not provide fully representative samples of the whole GC population, therefore the relative numbers of the R- and K-group cells is not conclusive. Note that non-labeled control cells were recorded in the original experiments from virus-labeled animals as well. However, in the two experimental arrangements these cells were recorded from different regions of the stratum granulosum, usually from the upper half in virus-injected animals to assess the most matured cells as controls and from the lower half in non-labeled animals. The description of these data in the revised Results reads as follows:

“Importantly, independent experiments, in which granule cells were recorded in non-labeled adult animals (see Methods), confirmed the coexistence of R- and K-functionalities among granule cells based on their ASL and VAR, and these individual cells showed similar correlations (or lack thereof) between their integrative functions and input resistance (Figure 4—figure supplement 1) as in the case of the above birth-dated data set (Figure 4).”

Concerns regarding “mechanism”: In the above new experiments we addressed the potential underlying mechanisms of the emergence of the two stable states by employing pharmacological or physiological interventions, which disrupt the presumed balance, which maintain two functionally stable states. The results are consistent with our original suggestion that a complex interaction of multiple mechanisms are behind the functional segregation between the two stable states. The first evidence for this is provided by the experiments, in which the biophysical and integrative properties of the same granule cells of non-labeled adult rats were assessed both at physiological and lowered temperatures (28-29ºC). In this situation the same mechanisms are available (such as voltage-gated channels), yet, the lower temperature affected the integrative properties of the cells depending on their functional states. Specifically, the average slope of R-group cells (with low initial value) tended to increase; whereas, in case of K-cells the initially high slope value decreased (Figure 4—figure supplement 2), as evidenced by the linear correlations. Thus, at 28-29 ºC the slope of all granule cells became similar regardless of their functionality at physiological temperature.

Interestingly, elevation of extracellular calcium concentrations from 2 to 4 mM (Figure 4—figure supplement 2), in order to proportionally increase the influence of the calcium-dependent mechanisms (such as after-depolarization or calcium-activated potassium conductances), was accompanied by a number of substantial changes in the excitability of the cells, including decreased ability to spike at high frequency, but did not disrupt the segregation between R- and K-group properties. In contrast, application of ML297, which activates GIRK channels, a known regulator of cellular excitability, specifically increased the slope in K-group like cells without affecting R-group cells (Figure 4—figure supplement 2). Thus, GIRK activation state-dependently affected the slope of the tested cells, similarly to the lower temperature, even though the two interventions has largely opposite effects on the classical measures of cellular excitability. Note that we did not detect state-specific effects of the above interventions on the classical biophysical parameters (e.g. Rin is similarly changed in R- and K-cells during GIRK activation) indicating again that a single parameter is not sufficient to determine the properties of input-output transformation of granule cells. Please refer to the new Figure 4—figure supplement 2 and the corresponding text in the Results read as follows:

“This latter result shows that consideration of multiple parameters by their arithmetical values is not sufficient to predict the functional separation of R- and K-groups. Thus, a complex and balanced interaction is probably behind the formation of the two stable states, as also suggested by additional experiments, in which the cellular excitability was challenged by pharmacological and biophysical interventions (Figure 4—figure supplement 2).”

The reviewers also state that *“there are obvious candidates (e.g.,* Ca^2+^
*channels, Schmidt-Hieber et al., Nature 2004)”.* Please note that in our sample we avoided the very young ABGCs (<3 weeks), which are not able to reliably generate high frequency sodium spikes and have a prominent low-threshold calcium channel contribution to their somatically recorded regenerative events as in the mentioned article. We do not show these preliminary recordings in our short paper but these results are in full agreement with previous publications showing the incapability of <3 weeks neurons in regards of full sodium spikes, probably one of the last components that is acquired by young ABGCs to become functionally completely integrated into the hippocampal network. Nevertheless, the experiments in which the integrative properties of individual cells were tested in two different calcium concentrations (see above) indicate that calcium channels are unlikely to directly contribute to the segregation between R- and K- functionalities.

*4) The authors assert that the two types of input-output transformations differentially support pattern separation at the cellular level. The idea that pattern separation (a network function) can occur on a single cell level requires further explanation or justification*.

We understand the reviewers’ concern and have clarified this point in our revised manuscript. We now state more explicitly how the distinct integrative properties of individual granule cells may contribute to different functions. We removed the unclear sentences regarding cellular pattern separation. The new Discussion section reads as follows:

“Thus, in the theoretical case of identical input strengths and patterns to R- and K-cells (which allowed us to investigate the integrative properties of individual ABGCs in isolation), sensitivity of ABGCs in the R-group to certain input ranges, in opposed to linear reporting in cells of the K-group, allows a certain level of input disambiguation at the level of single granule cells.”

*5) The authors used running wheels/exercise that has been shown to accelerate rate of maturation of adult-born dg neurons*. *As such, this experimental design confounds the extrapolation of “stage specific” properties at baseline. The authors must explicitly acknowledge this potential confound and the possibility that the window of the R and K stages may actually be shifted farther to the right in the lineage at baseline (without exercise induced acceleration of maturation). How do the authors rule out the possibility that the distribution of groups (R, K) may be due to the differential effects of modulatory afferents onto maturing ABGCs by running?*

We fully agree with the reviewer that that experience may change the cellular behavior and functionality of newborn granule cells that may result in experience-dependent occurrence of R- and K-group properties.

Therefore, to avoid any influence of the different properties by the environment we used the same environment throughout the study, including the new experiments shown in the figure supplements (see points 2, 3, and 6).

Nevertheless, the larger housing cage with running-wheel probably represents an environment, which is closer to the natural environment of the animals than the standard animal housing conditions. Furthermore, a technical advantage of the running wheel environment is that it is known to lead to an increased number of ABGCs during the studied post–mitotic periods. However, we have now added two new sections explicitly discussing the potential influence of experience on the emergence of K and R groups. These sections read as:

(Discussion):

“Moreover, environmental conditions that increase or reduce hippocampal neurogenesis may have specific effects on the relative contribution of newly generated granule cells to the R- or K-groups that may explain distinct behavioral consequences of altered neurogenesis (8; 11; 31).”

Methods: “After the surgical procedure two or three siblings were housed together in large cages (75 cm x 35 cm) equipped with a running wheel, toys and shelters until the electrophysiological experiments because running is known to increase the number of surviving adult-born neurons (11; 29; 7).”

Certainly, future experiments will be performed in which the parameters of environment and behavior are fully controlled in multiple directions in addition to the age of individual cells. However, we are convinced that our data showing for the first time the functional segregation of age-matched newborn granule cells in the adult dentate gyrus will set the ground for these future experiments that currently go beyond the scope of our study.

Concerns regarding “modulatory afferents”: The Results suggest that a more direct and tonic activation of conductances by modulatory afferent is unlikely to contribute to the segregation of R- and K-functions because such effect should have been detected on the passive properties of the cells (e.g. Rin, membrane potential). On the other hand, the experiments with synaptic receptor blockers (see point 2) exclude the possibility that the different gain of the input-output function is due to group-specific differences of indirectly activated feedback synaptic inputs (e.g. mossy cells, or GABAergic cells).

*6) It is not clear why such young rats were used for this study (besides the fact that this age may be more amenable for retroviral injections)*, *if the authors were interested in studying adult hippocampal neurogenesis? Do we know that the same distributions of ABGCs will be found in adult brains?*

We have performed new experiments to address the potential age dependence of the presence of the R and K-functionalities. Thus, we tested the integrative properties of granule cells in P17-18 rats (note this is the age of the animal and not the postmitotic age of the tested cells), in which all spiking granule cells are from the first generation of developmentally generated population. Surprisingly, the tested cells were not separable based on the slope of the input-output function and we found that large majority of the tested cells behaved as a K-functionality (see the Figure below showing this data) similar to the oldest ABGCs (10 weeks old). This is a potentially important observation because it may suggest that at the level of the slope of input-output function developmentally born cells mature differently than adult-borns.

Importantly, when all cells from these young animals are considered, their slope of input-output function showed a very similar correlation to their input resistance as in the case of K-group cells from adult animals (1.93±0.29/GΩ in P17-18 vs. 1.99±0.29/GΩ K-cells in adults). This correlation holds in spite of the larger average and wider range of input resistance (288±23MΩ vs. 191±10MΩ of the cells in young animals compared to K-cells in adults. This shows that in young animals practically all granule cells follow the generally accepted dependence of the slope of input-output curve on the general parameters of cellular excitability such as input resistance; however, this is true only to a fraction of adult-born granule cells, the K-group members. In young animals we used the same analysis criteria as in the recordings from adult animals and the cells were usually recorded in the lower half of the granule cell layer, as in the adult sample.Author response image 1.

In addition, to provide a good control for the main experiments of this manuscript by showing the above correlations, these control experiments raise several questions that are not the focus of the current study. For example, it raises the possibility that at the level of specific intrinsic cellular properties the developmentally generated and adult-born granule cells may mature differentially and the R-functionality is specific to ABGCs. However, we refrain to make such a fundamental and strong statement based on this single observation, which is not directly designed to address this question.

In full agreement with the literature we found that the age of animals plays a role in the rate of neurogenesis or the survival of new neurons. Therefore, for the birth-dating experiments we chose an age where there is still high number of ABGCs can be reliably detected by the virus labeling. We cannot exclude the possibility that at other ages the maturation may differ, especially, in young animals when the network itself is developing as well, which is less prominent in P30 or older animals. However, it is also important to note that the recordings of birth-dated ABGCs were performed in adult rats with the age > 7 weeks. The rationale of using P31-33 old rats for the virus-labeling and the potential age dependency of the neurogenesis and maturation is highlighted in the revised manuscript. This section reads as follows:

“Animals of this age were used because relatively large number of labeled ABGCs can be analyzed at a time of recording when the network properties of the dentate gyrus circuitry can be considered adult (14).”

*7) Because 5-9 weeks old neurons are made up of R and K like cells, how might this mixed population contribute to encoding functions (pattern separation and completion) of the DG. In other words, a greater discussion on the significance of variable I-O relationships in classmate cells to encoding functions is warranted*.

We agree with the reviewer and have revised the second paragraph of the Discussion to specifically highlight that the two modes of output rate generation within the functionally mixed population of 5-9 weeks old ABGCs reflect distinct aspects of their inputs. Note that the synaptic output of granule cells onto CA3 principal cells is suitable to faithfully reflect the firing rates of granule cells due to its short-term facilitation. The modified section reads as follows:

“Around the third postmitotic week, ABGCs represent a functionally homogeneous population (R-group) characterized by highly variable and sensitive output, which potentially underlies the effective disambiguation of input patterns because the output of R-group cells represent certain input ranges with exceptional efficacy.

Strikingly, our data indicate that “classmate” cells (born during the same period) can contribute to the network with fundamentally different functions during an extended period after cells are born (5-9 weeks). Conversely, these data suggest that similar cellular functions can be served by ABGCs that were born at different periods during the animal’s life. Therefore, the properties of individual cell rather than the cells’ age determine how certain input strengths are computed; either by, disambiguating certain input combinations (R-functionality) or by representing all of its inputs by similar rate changes toward the downstream networks (K-functionality).”

*8) The Discussion section is confusing and needs revision to accurately reflect the actual findings of the experiments. For example, the last two paragraphs of the Discussion are almost disconnected. In paragraph 1, the authors indicate that cells in the third post–mitotic week are “functionally homogeneous”, then in the next paragraph they indicate that “classmate” cells contribute to the network with “functionally different functions”*.

We understand this concern and have clarified the Discussion by being more specific with the age of the considered cells for certain functions. The modified section reads as follows:

“Here we show that the gain of the input-output transformation of 3-10 weeks old ABGCs exists at two functionally distinct states, allowing for the translation of similar excitatory drives into highly distinct action potential outputs in a manner that is not directly predicted by the cellular age alone (refer to the alternative maturation hypotheses depicted in Figure 1). This finding is in contrast to the continuous maturation hypothesis (Figure 1).”

Discussion section: “Strikingly, our data indicate that “classmate” cells (born during the same period) can contribute to the network with fundamentally different functions during an extended period after cells are born (5-9 weeks). Conversely, these data suggest that similar cellular functions can be served by ABGCs that were born at different periods during the animal’s life.”

*9) The word “stable” in the title does not seem to be supported by the data*.

We following the reviewers’ advice and have changed the title accordingly.